# Main Husbandry Practices and Health Conditions That Affect Welfare in Calves: A Narrative Review

**DOI:** 10.3390/ani15213064

**Published:** 2025-10-22

**Authors:** Eva Mainau, Laurent Goby, Xavier Manteca

**Affiliations:** 1AWEC Advisors SL, Ed. Eureka, Parc de Recerca de la UAB, Bellaterra, 08193 Barcelona, Spain; 2Boehringer Ingelheim Vetmedica GmbH, Binger Str. 173, 55216 Ingelheim am Rhein, Germany; laurent.goby@boehringer-ingelheim.com; 3Department of Animal and Food Science, Autonomous University of Barcelona, Bellaterra, 08193 Barcelona, Spain; xavier.manteca@uab.cat

**Keywords:** calves, welfare, health, behaviour, management

## Abstract

**Simple Summary:**

The early life of a calf is a critical time that greatly affects its health and welfare. How calves are born, fed, housed, and cared for during this period plays a major role in how they grow and develop. This review looks at the main ways that farming practices impact young calves, such as how they are offered milk and water, and how they are weaned, housed, and transported. It also offers clear recommendations to help farmers improve calf care. Good practices include giving calves enough high-quality colostrum (the first milk), providing fresh water from birth, feeding the right amount of milk, and weaning them gradually. Keeping calves in small social groups and using proper pain relief when removing horns or castrating are also important. Clean environments and preventing the spread of disease are key. However, there are still areas where more research is needed, including how to measure milk quality on farms, how to wean calves in different systems, and the long-term effects of pain in early life. By improving these practices, we can help calves live healthier, more comfortable lives and support responsible, ethical farming.

**Abstract:**

Calf welfare is critically influenced by early-life husbandry practices and health conditions. This narrative review synthesizes current evidence on key management practices affecting calf welfare, including calving, colostrum intake, milk feeding, water provision, weaning, housing, mutilations, and transport. A structured literature search was conducted in Web of Science and Scopus using general and topic-specific keywords, complemented by expert opinions from EFSA. Evidence-based recommendations are presented to improve calf welfare, emphasizing timely colostrum administration, biologically appropriate milk volumes, access to clean water from birth, gradual weaning, and stable social housing. Pain mitigation during disbudding and castration, along with strict biosecurity and hygiene, are essential to reduce disease risk. Despite advancements, significant knowledge gaps persist, including practical tools for on-farm colostrum assessment, optimal weaning protocols, the long-term impacts of early-life pain, and alternatives to current transport practices. The review highlights the need for standardized protocols, validated technologies, and enhanced training for farmers and veterinarians. Improving husbandry practices based on scientific evidence is essential to enhance calf health, productivity, and ethical sustainability in modern rearing systems.

## 1. Introduction

It is generally accepted that animal welfare comprises physical and mental health [1]. Animal welfare includes several aspects, such as the absence of thirst, hunger, discomfort, disease, pain, injuries, and stress, as well as the possibility of expressing normal behaviours [2]. Recently, the primary focus has shifted from unpleasant experiences only to also include pleasant ones, as well as their long-term effect. Welfare is now seen as the balance between pleasant and unpleasant experiences over time, rather than just the absence of suffering [3].

Fraser and Broom [4] documented the link between welfare, animal behaviour, and health. Animals that are cared for in accordance with acceptable welfare standards are more likely to be healthy and, conversely, animals kept in poor welfare conditions are often at a greater risk of disease. That is because improved welfare can be a means to improve immune function, including response to treatment, vaccines, or infection. However, the relationship between welfare, immunity, and disease resistance is more complex than it appears [5]. Immune responses operate through two primary mechanisms—the innate and the adaptive responses—each with distinct implications for animal welfare. The innate immune response serves as the first line of defence against infection or injury. It is non-specific and rapidly activated, involving the release of pro-inflammatory cytokines and the induction of local inflammation. These processes often result in systemic effects such as fever, anorexia, and lethargy—collectively referred to as “sickness behaviours.” Although these responses are adaptive and facilitate recovery, they temporarily compromise animal welfare by reducing comfort and normal functioning. In contrast, the adaptive immune response is antigen-specific and characterized by the production of antibodies and the development of immunological memory. Activation of this system requires substantial metabolic resources, particularly during the initial phase. However, it contributes positively to long-term welfare by reducing the severity and recurrence of disease through enhanced immune protection [5,6].

Understanding the physiological and behavioural consequences of immune activation is essential for evaluating and improving animal welfare, particularly in contexts involving disease control and husbandry practices.

Animal husbandry practices significantly influence both animal welfare and the incidence of disease. Conditions arising from physiological imbalances—whether or not they involve infectious agents—are closely linked to husbandry practices and may strongly interact with the overall welfare status of the animals [7].

On intensive farms, calves are exposed to a series of challenges from birth, including difficult calvings, early separation from their mothers, and housing in individual hutches until they are weaned at an early age and transferred to large groups. During this period, there are many factors that can compromise calf welfare, including nutrition, comfort, health, and their emotional state [8,9,10]. According to the EFSA Scientific Opinion on Calf Welfare [11], the most common welfare issues in UE calf-rearing systems are health problems (such as respiratory and gastrointestinal disorders) and husbandry-related issues such as limited exploratory behaviour and stress from group housing. However, there are limited published data on husbandry systems and practices that improve animal welfare.

In fact, reviews have focused on calving (e.g., [12,13,14]), perinatal mortality in calves (e.g., [15,16]), or both (e.g., [17,18,19]). An extensive number of reviews have addressed colostrum management (e.g., [20,21,22,23]), and/or milk feeding (e.g., [24,25]), housing (e.g., [26,27,28,29]), and health (e.g., [30,31]). However, reviews summarizing the most important aspects of management practices and health issues in calves in relation to animal welfare are scarce or outdated ([32], published in 2008). Since 2008, several new or increasingly recognized animal welfare issues have emerged in calves, such as cow–calf contact versus early separation and the transportation of pre-weaned calves. Also, advances in technologies, including automatic milk feeders and digital monitoring tools, have appeared or become more widely used. This narrative review aims to describe the impacts of various husbandry practices and health conditions on calf welfare in a single comprehensive document. Resulting from this review, evidence-based preventive measures and practical recommendations will be proposed, and knowledge gaps in this area will be identified.

## 2. Materials and Methods

All searches were conducted using the general keyword search terms ((calf OR calves) AND (welfare OR wellbeing OR well-being) OR (management OR (health OR disease)) in Web of Science and Scopus. Searches were restricted to title, abstract, and keywords of all peer-reviewed scientific articles and books published from 1990 to August 2025. Additionally, expert opinions from the European Food Safety Authority (EFSA) has also been included. The general search terms were combined with specific keywords for each topics (and subtopics): assisted calving (including calving, perinatal mortality and cow–calf separation), colostrum management, milk feeding management, water provision, weaning management, housing (including social housing, thermal stress and stocking density), diseases (including diarrhea, respiratory disorders, omphalitis, and arthritis), husbandry painful procedures (including disbudding or dehorning and castration), and transport. Each topic and its subtopic were searched separately. Table 1 shows the general and specific keywords used, as well as the inclusion and exclusion criteria for databases searches. References were organized using the “Deduplicator” and “Screenatron” functions of the Evidence Review Accelerator (https://tera-tools.com/, accessed on 1 July 2025) and managed for citation purposes with Zotero software (version 7.0.24, Corporation for Digital Scholarship, Vienna, VA, USA).

## 3. Assisted Calving

Dystocia is defined as calving difficulty resulting from prolonged parturition or assisted extraction [12] and is associated with high levels of pain in both the mother and the calf [13]. In dystocic calvings, calves experience pain due to traumatic injuries (e.g., fractured leg) or prolonged or forceful traction suffered during and immediately after calving (e.g., mechanical calving aid) [33]. Dystocia increases the risk of stillbirth [12,34] and perinatal mortality [34,35], which are indicators of impaired health and welfare [36,37]. On dairy farms, perinatal mortality ranges from 2 to 10% [15,38,39], with the majority of deaths (75%) occurring within the first hour after birth [18].

Dystocia also has an effect on neonatal vitality and subsequent health. Dystocic calves often exhibit delayed sternal recumbency and slower standing due to hypoxia in the birth canal. This delay can impair colostrum intake and/or immunoglobulin (Ig) absorption, increasing the likelihood of failure of passive transfer (FPT) [40,41,42]. Consequently, FPT is associated with elevated risk for mortality (OR = 2.12), bovine respiratory disease (OR = 1.75), diarrhea (OR = 1.51), and overall morbidity (OR = 1.91), highlighting the cascading and long-lasting consequences of dystocia on calf health [43].

### 3.1. Practical Recommendations on Assisted Calving

Recommendations for veterinarians and farmers to ensure successful calving in cows include preventive measures during late gestation and earlier, protocols for appropriate intervention during calving, and proper management after calving.

During late gestation, it is recommended to provide a balanced diet and avoid overfeeding to prevent dystocia [35,44]. The implementation of measures aiming to avoid heat stress are crucial, as heat stress in late gestation may lead to FPT regardless of colostrum quality [45,46]. Additional effort should be made to correct modifiable risk factors of dystocia, such as age at first freshening, dry period length, or sire choice [18].

It is fundamental to provide cows with a particular infrastructure for calving [47]. Proudfoot [48] concluded that the ideal maternity pen should allow for the expression of a cow’s natural maternal behaviour, including the ability to seek an isolated space before calving. In pen group calvings, it may be beneficial to provide access to a barrier or some type of hiding space where the cow can seek partial seclusion from her herd mates [48].

Training farm personnel to recognize and manage difficult births is essential for reducing perinatal mortality [17,49]. The cow should be monitored once every hour from the onset of the first stage of calving [50]. Intervention is only necessary if any stage of the calving process is excessively protracted and/or if atypical behaviour, or normal behaviour with abnormal frequencies, is observed. In farms where constant calving surveillance is not guaranteed, the use of automatic devices for calving prediction could provide an alternative [51,52]. After calving, dams should be allowed to lick and ingest the amniotic fluid on the calf. This behaviour not only increases the calf ’s vigour, but also helps to reduce the dam’s post-calving pain, as the amniotic fluid contains compounds that enhance the analgesic effect of endogenous opioids [53].

The timing of cow–calf separation time after calving has several consequences for calves [14]. When calves are left with the cow there is a reduction in cross-sucking behaviour [54], an increase in play [55], and improved average daily gain (ADG) [11,54,56]. However, separation after the mother–calf bond develops has negative consequences for the cows (e.g., vocalizations) and calves in terms of behaviour and emotional state [56,57,58]. In consequence, EFSA [11] recommended that the calf should be kept with the dam for around 24 h (before the bond is formed) and be housed with another calf after that. If the calf is left with the dam, a well-designed programme of “assisted nursing” should be considered [59].

### 3.2. Main Research Gaps on Assisted Calving

Definitions of perinatal mortality vary considerably, particularly regarding the inclusion of stillbirths and neonatal deaths within the first 48 h postpartum. This lack of standardization poses a significant challenge to the comparability of international datasets and hinders the development of universally applicable benchmarks [16].

Potential solutions involve developing and adopting internationally agreed definitions for perinatal mortality, implement standardized record-keeping systems at the farm level, and providing training to farmers and veterinarians on consistent data collection and reporting.

Further investigations into the effects of management practices (e.g., supervision, assistance protocols) and environmental conditions (e.g., temperature, bedding) during calving time on cow calving behaviour are needed [47]. Potential solutions should focus on conducting controlled studies to identify optimal calving conditions (e.g., space, bedding, supervision). In addition, developing guidelines for best management practices at calving based on scientific evidence and promoting the use of technology to detect problems early should be considered.

Additional research is needed on cow–calf separation time to provide sufficient evidence on the benefits and risk factors associated with the welfare of calves and cows. Potential options include performing comparative studies on early vs. delayed separation to identify optimal practices, as well as exploring alternatives to abrupt separation (e.g., fence line contact systems). It is essential to develop scientifically grounded, welfare-oriented recommendations and effectively communicate the findings to farmers.

## 4. Colostrum Management

Colostrum management is the single most important factor influencing calf health and survival [21]. Newborn calves are deficient in serum Igs levels, and consequently entirely dependent on intake from maternal colostrum [60]. Colostrum provides Igs (IgG, IgA, and IgM), which are essential for the transfer of passive immunity, along with various immune cells (lymphocytes, macrophages, and neutrophils) and a highly digestible source of energy and protein [16]. The inadequate ingestion of colostrum and insufficient transfer of Igs into the calf’s bloodstream are defined as failed transfer of passive immunity (FTPI). Calves with FTPI are more susceptible to enteric and respiratory diseases [11,61] and face a higher risk of calf mortality [62,63].

Several reviews have highlighted the main factors that determine the successful transfer of maternal immunity to calves [19,21,23,24,64]. These include the quality of colostrum, measured by IgG concentration; the quantity ingested; and the calf’s ability to absorb Igs in the small intestine. Other factors, such as colostrum origin, hygiene, storage conditions, and how long calves continue to receive colostrum after birth, have also influence calf health and mortality risk.

Colostrum quality is often assessed with a Brix refractometer, which estimates total serum protein (TSP). This method is reliable because TSP strongly correlates with IgG concentrations in the blood during the first days of life [65]. Several studies have determined cut-off points for passive immunity transfer in calf serum according to TSP and IgG concentration and the equivalent measured with a Brix refractometer (see Table 2 in [19] for details).

Colostrum quantity is equally important. To achieve the successful transfer of passive immunity, it is commonly recommended to feed 3 to 4 L of colostrum, which corresponds to 10–12% of the body weight (BW) of a newborn calf [20,66]. However, few studies provide evidence to support this guideline [67]. Feeding large volumes of colostrum (e.g., ≥4 L) at once may lower IgG absorption due to mechanical abomasum distension and delayed emptying [68]. Slower abomasal emptying may, in turn, reduce the absorption of colostrum components [69]. A second feeding after the first colostrum meal may be important, as it has been shown to reduce FTPI and morbidity [70]. Additionally, prolonged colostrum feeding and/or feeding transition milk (milkings 2 to 6 after calving; [71]) have been demonstrated to result in higher BW at weaning and improved health [71,72].

Finally, the timing of colostrum administration is critical. The absorption of Igs in calves is optimal within the first 4 h after birth, declines rapidly after 12 h, and ceases approximately 24 h postpartum [73].

Colostrum with high bacterial counts has been shown to impair IgG absorption [74]. This is a concern because pathogenic bacteria in colostrum can interfere with Ig absorption and may also cause diseases such as diarrhea or septicaemia [20,75].

### 4.1. Practical Recommendations on Colostrum Management

All calves should consume a volume of high-quality colostrum equivalent to 10–12% of their BW in the first 2–3 h after birth (e.g., 4 L for a 40 kg calf), followed by an additional meal equivalent to 5% of their BW 6–12 h later [19,21]. Following the initial colostrum intake, calves should be provided with transition milk or a mixture of milk and colostrum for at least 4 days. Milking the cow as soon as possible after calving is highly encouraged and will likely result in the highest colostral IgG concentrations [76].

It is recommended to monitor the quality of the colostrum using a refractometer or a colostrum metre. In general, it is assumed that high-quality colostrum has an STP > 52 g/L, or an IgG concentration > 25 g/L, or a Brix value > 9.4% [19]. Colostrum should ideally be sourced from the calf’s mother or another cow from the herd, as it contains antibodies specific to the farm’s environment. Colostrum with high microbial count (>100,000 cfu/mL) or high coliform counts (>10,000 cfu/mL) must not be fed [77,78]. Fresh colostrum could be pasteurized for 30 min at 60 °C in order to minimize the amount of pathogens in colostrum [76]. Feeding equipment (teats, bottles, buckets, stomach tubes) should be thoroughly cleaned and disinfected after each use and between calves. If colostrum is not administered immediately, it is recommended to refrigerate it at 4 °C [79] for up to 48 h or freeze it at −18 to −25 °C for up to 1 year. Frozen colostrum should be thawed in a water bath at 38 to 40 °C (never microwaved) to minimize IgG loss [76].

If a newborn calf does not suck colostrum from the cow or a bottle (e.g., due to weakness, illness, or lack of interest) a clean oesophageal or stomach tube can be used to facilitate direct feeding [76].

### 4.2. Main Research Gaps on Colostrum Management

Reliable on-farm evaluation of colostrum quality is critical for ensuring adequate passive immunity in calves and reducing morbidity and mortality. Further research is needed to validate more practical and cost-effective tests for assessing colostrum quality at the farm level.

Different studies use varying thresholds of good-quality colostrum. This lack of standardization can lead to inconsistent management practices and suboptimal calf health. Possible solutions include conducting longitudinal studies to correlate different colostrum quality cut-offs with calf health and growth.

While the concentration of immunoglobulin IgG has been the primary focus in this field, the effects of colostrum management on colostral leukocytes and other immune compounds remain largely understudied, despite their potential importance for calf immunity and development [76]. Potential solutions include conducting research to quantity and understand the role of colostral leukocytes and other immune compounds, evaluating how colostrum management practices (e.g., timing or storage) affect these components, and integrating these findings into comprehensive colostrum management protocols that optimize overall immune transfer, not just IgG.

## 5. Milk Feeding Management

Traditionally, sucking calves are fed a daily amount of milk or milk replacer equivalent to 10% of their BW (i.e., 4 L–6 L of milk per day) to promote starter concentrate intake and allow early weaning [80]. However, this practice can negatively affect welfare, growth, and long-term productivity. Feeding calves at 10% of BW has been linked to increased frustration and hunger-related behaviours [19,26], including vocalizations (e.g., [81]), increased searching for milk or unrewarded visits to the milk feeder (e.g., [82]), excessive sucking of penmates or equipment [83], and reduced play behaviour [84,85]. In consequence, it is currently recommended to feed calves a milk allowance of 20% of BW, which aligns with a biologically normal milk intake [80,86]. This practice enhances preweaning growth and future lactation performance [87,88,89]. Ellingsen et al. [90] showed, using radiographs, that the abomasum has a high capacity for distention. Voluntary meals of 2.5 L to 6.8 L of milk given by teat to 3-week-old calves did not enter the rumen, nor were any behaviours indicative of abdominal pain or discomfort observed.

In addition to milk quantity, feeding frequency and the method of milk administration also influence calf health and behaviour.

Regarding frequency, calves that have free access to their dam suckle 4–9 times a day [91,92]. The frequency of suckling declines with calf age, e.g., from 6.3 times/24 h at 2 weeks of age to 3.8 times/24 h at 8 weeks of age [91], while suckling duration per session increases, e.g., from 6.2 min at 2 weeks of age to 8.8 min at 4 weeks of age [93]. Restricting feeding frequency to only once per day may not meet the calves’ energy needs for optimal growth and welfare. Indeed, glucose concentrations remain fairly stable for 12–14 h after milk ingestion, but drop significantly after 16 h, leading to energy deficits [94].

Regarding the method of milk administration, feeding calves with a nipple system (also called teat-feeding or bottle-feeding) instead of buckets offers several benefits for calf health and behaviour. Nipple feeding encourages slower milk intake, similar to the calf’s natural sucking behaviour when nursing from its mother. This method allows milk to bypass the rumen and reach the abomasum directly by more effectively closing the oesophageal groove [95]. As a result, milk is less likely to ferment in the rumen, reducing the risks of bloating, acidosis, and diarrhea [96]. A number of other factors trigger this oesophageal reflex, including warm milk and the position of the calf’s head while drinking [97]. The natural suckling action also stimulates saliva production, which provides digestive hormone secretion enhancing digestion and nutrient absorption [98]. By contrast, bucket-fed calves are more likely to develop stress-related behaviours, such as abnormal oral activities directed toward pen fixtures or cross-sucking on other calves’ ears or navels, which can lead to injuries and infections [83,99].

The milk fed to calves can be either a milk replacer, whole milk from the bulk tank, or waste milk (e.g., milk from cows with mastitis [100] or from cows treated with antimicrobials [101]. Nutritional diarrhea typically results from the use of poor-quality milk replacers and/or management mistakes, such as the incorrect calculation of milk replacer concentration [102,103,104]. Poor-quality milk replacers also appear as a risk factor for respiratory problems: Medrano-Galarza et al. [105] reported that feeding whole milk instead of a milk replacer was associated with a lower prevalence of respiratory disease. Milk replacer must be of high quality, as poor-quality milk replacers can affect welfare through diarrhea morbidity and also hunger through starvation [106]. It is essential to ensure the protein content remains above 28% [107,108], as this is directly related to daily gain [109]. Fat content must be maintained in the range of 17–25% [72,107,110]. Waste milk can vary nutritionally [111,112] and may increases the transmission of infectious pathogens and antibiotic resistance [10,113,114]. Pathogens can be shed in the milk of infected cows and cause diarrhea, pneumonia, otitis media, or arthritis in calves [115,116].

Regardless of the milk-feeding strategy adopted, the correct implementation of hygiene is essential to prevent health problems in calves, reduce the burden of pathogenic bacteria, and break the chains of infection. Several studies specifically highlight the cleaning of artificial nipples and buckets, identifying these components as key critical points [117].

Finally, several studies reported decreased occurrence of diarrhea and reduced days with diarrhea when dairy calves were fed a biologically normal amount of milk [88,118]. Such health benefits may be attributed to the positive effects of nutrition on immune function [119], which in turn increases tolerance to infections post-weaning [24,120]. Conversely, some studies have shown that calves fed a biologically normal milk intake have higher fecal scores, reflecting softer or more liquid feces and, consequently lower fecal consistency [121,122,123,124]. However, Liang et al. [123] reported that fecal consistency may not be the most reliable indicator of enteric health.

### 5.1. Practical Recommendations on Milk Feeding Management

Restricted feeding of calves (10% BW) is not recommended as it leads to chronic hunger, malnutrition, and immunosuppression [125]. In contrast, ad libitum milk feeding supports better growth but may hinder rumen development due to reduced intake of solid feed [126,127]. Consequently, it appears that providing milk at a level equivalent to 20% BW should be recommended during the first 4 weeks of life. This approach may help reduce negative effects on calf welfare, even though it might limit solid feed intake and rumen development prior to weaning [24,128].

The optimal number of dosed meals per day remains unclear. However, at least 2 meals per day are recommended to satisfy both feeding motivation and nutrition for growth [129].

Feeding calves using a nipple system rather than buckets is recommended, as it offers several benefits for calf health and behaviour. However, special attention should be paid to hygiene [117] and the maintenance of teats (e.g., hole size).

The type of milk provided to calves has not been shown to produce significant differences in their welfare and developmental outcomes. It is recommended to provide whole milk or high-quality milk replacer, maintaining a protein content above 28% [107,108] and a fat range of 17–25%. It is suggested not to feed calves with waste milk, due to the possibility of the development of antimicrobial-resistant bacteria and the variability of the daily nutritional composition of waste milk.

### 5.2. Main Research Gaps on Milk Feeding Management

The review reveals a notable absence of a clearly defined protocol concerning the quantity, concentration, and frequency of administration (see revision [19] for further information). To address this gap, controlled studies are warranted to elucidate these parameters, thereby enabling the development of standardized and evidence-based guidelines for milk administration.

## 6. Water Provision

Water availability is essential for animal welfare, as it satisfies thirst and directly influences feed intake and BW gain [130,131]. Drinking water from the early days of life may aid in fibre intake and possibly rumen development [132]. Water goes directly into the rumen, creating optimal conditions for fermentation by supporting the activity of rumen bacteria [133,134]. By contrast, milk goes directly into the abomasum via the oesophageal groove and does not contribute to the development of the rumen [133].

Survey data from Denmark, Norway, the USA, Chile, and Canada (Jensen and Vestergaard [135] indicate that a high percentage of calves were not provided with continuous access to water. One reason producers may be reluctant to offer water to newborn calves is the belief that this could cause diarrhea [135,136]. However, Wickramasinghe et al. [136] confirmed that the age at which drinking water was offered to calves (from birth or from 17 d after birth) did not have an effect on the severity of diarrhea or the number of days that calves had diarrhea.

In relation to water quality, the most important physicochemical parameters to take into consideration are pH, total dissolved solids, hardness, excessive amounts of minerals (such as nitrates, chloride, and sulphates), and microbial loads [137]. These parameters, affect calf health, immunity, morbidity, and various causes of calf mortality in cattle herds [138]. Kamal et al., [139] analyzing 132 Egyptian cattle farms suffering from emerging epidemics, concluded that the mortality rates of calves showed a moderate positive correlation with all physicochemical and microbial aspects except pH. Providing fresh and good-quality water also prevents the contamination of calves with agents such as *E. coli* and *Cryptosporidium* [11,140].

### 6.1. Practical Recommendations on Water Provision

Water availability and water quality are critical for calf welfare, influencing feed intake, rumen development, and growth. Calves should be given early access to clean and fresh water, even if they have free access to milk, because milk alone does not meet all their water requirements [11,141,142].

### 6.2. Main Research Gaps on Water Provision

Understanding of how unweaned calves consume water remains limited, particularly in relation to factors such as age, breed, environmental conditions, and diet. To address this gap, both observational and experimental studies are needed to evaluate variations in water intake across these variables. Additionally, research should explore the influence of management practices, such as group housing vs. individual pens, on water consumption. Furthermore, the influence of water accessibility (e.g., location, flow rate, and delivery method) warrants deeper investigation, as does the impact of water provision on the health and development of calves during the pre- and post-weaning periods.

## 7. Weaning Management

Under natural conditions, the weaning process is gradual, occurring over several weeks, and is often not completed until the calf is 8 to 11 months old [143]. In contrast, on commercial farms, weaning is usually abrupt, occurring over shorter periods of time (often less than a week), and calves are weaned at around 8 weeks of age [144].

Weaning is highly stressful for animals [8] and poses a significant challenge for farmers [10]. It involves nutritional, social, physical, and physiological stressors [145]. Weaning causes behavioural changes in calves, such as anxiety and frustration [19]. It is often accompanied by signs of stress and distress, including increased plasma cortisol concentrations [146], increased vocalizations [10,147], and signs of hunger and weight loss [148,149]. Furthermore, weaning results in a temporary reduction in immune function in calves immediately post-weaning, predisposing them to respiratory infections [145]. For instance, adopting best practices during weaning and avoiding early weaning can reduce the prevalence of bovine respiratory disease [150,151]. However, a systematic review by Welk et al. [152], which aimed to summarize the effects of weaning practices on calf health, concluded that very few studies have examined calf health in relation to different weaning methods, and it is not possible to extract a valid conclusion.

Various weaning strategies have been explored to promote rumen development and growth while reducing hunger and distress. These approaches include (1) delaying the age of weaning, (2) gradually reducing the amount of milk offered (step-down methods), and (3) encouraging greater solid feed consumption [19,153]. Such strategies are especially relevant for calves fed a high amount of milk. Late-weaned calves tend to adapt better to weaning, showing an increase in solid feed intake and weight gain during the transition period [154]. Similarly, step-down protocols used with high milk-feeding early in life stimulate solid feed intake, which supports gastrointestinal tract development, growth, and overall health [153]. Moreover, combining step-down weaning with criteria based on solid feed intake further enhances pre-weaning solid feed consumption. This approach helps prepare the gut for post-weaning digestion and improves nutrient utilization [153,155].

### 7.1. Practical Recommendations on Weaning Management

To minimize cumulative stress and its potential impact on calf welfare and development, weaning should be carefully timed to avoid concurrence with other stress-inducing procedures or environmental changes, such as re-grouping, relocation within the facility, disbudding, castration, clinical signs of illness, or transportation [145].

A gradual reduction in the amount of milk (step-down method) is recommended from 40 days of age, as starting earlier may impair growth and cause malnutrition [155,156]. This process should last at least 10 days [157], with progressively limited milk access.

Weaning should be based on solid feed intake: calves must consume at least 1 kg of solid feed per day for three consecutive days [158] or a cumulative total of 15 kg of non-fibre carbohydrates [159]. This typically occurs by 8–9 weeks of age, and weaning should not begin before then. High-milk diets may delay weaning to around 12 weeks. Automatic milk feeders support gradual daily milk reduction [153] and allow for individual weaning [160,161].

### 7.2. Main Research Gaps on Weaning Management

Additional research is needed to determine the optimal weaning age for different systems to minimize stress and maximize growth. The ideal age for weaning calves can vary depending on several factors, such as breed, environment, and management practices. Potential solutions include conducting controlled studies to identify key factors and establishing standardized protocols for the weaning process.

With the advancement of monitoring tools and technologies, more research is needed on implementing and validating sensors or other technology to monitor calves during the weaning process to track stress levels, health, and growth in real-time.

## 8. Housing

### 8.1. Social Housing

Research shows that calves housed with social contact—either in pairs or groups—experience better welfare compared to those housed individually. Social housing supports the development of social and cognitive skills, which are essential for handling new situations later in life [9,26,162]. It also improves productivity: socially housed calves eat more solid feed, grow faster (higher ADG before and after weaning), and reach a greater weaning weight [29]. These benefits are partly due to reduced food neophobia—the hesitation to try unfamiliar foods—thanks to early social experiences [163,164]. However, negative behaviours such as cross-sucking, competition, or aggression can also occur among grouped calves [165,166]. Strategies to prevent these issues have been reviewed by Costa et al. ([26]) and are summarized in the Recommendations section of this manuscript.

Studies on the impact of social housing on calf health show mixed results. Some studies indicates that group-housed calves experience higher rates of illness (e.g., [167,168,169]) and mortality (e.g., [170]) compared to individually housed calves. However, other studies report no increased health risks (e.g., [147,171]) or even health benefits from group housing [172,173]. Therefore, the increased occurrence of health issues associated with the type of housing may be influenced by other factors. These conflicting findings suggest that health outcomes may be influenced by additional factors—such as nutrition, management practices, and environmental conditions—rather than housing type alone [150,169].

To date, two key managements factors have been identified as critical for achieving the benefits of social housing while minimizing the risk of health disorders such as respiratory problems and/or diarrhea. The first factor is an appropriate group size, which will allow calves to experience the welfare benefits of contact with peers without increasing the risk of infectious diseases [11]. According to an EFSA scientific report [11], the prevalence of respiratory disorders in calves housed in groups of 2–3 animals is similar to that in individually housed calves and in groups of 4–7 calves. However, the prevalence was considerably higher in groups of 12–18 calves and in groups of 30–40 calves. The second factor is the stability of group composition, as the prevalence of both diarrhea and respiratory disease is more than twice as high in dynamic groups compared to stable ones [170,174]. Frequent changes in group composition causes stress, which can impair immune function and increase susceptibility to infection [151,175]. Finally, the impact of age at grouping on calf welfare remains unclear and requires further investigation. Most studies recommend establishing groups between 3 and 6 days of age [11]. So far, no significant differences have been found in terms of solid feed intake, growth [176], or social behaviour [177] between calves paired immediately after birth and those grouped at 3 weeks of age.

### 8.2. Thermal Stress

Thermal comfort in animals is measured by the thermo-neutral zone, which is the range of ambient temperatures in which an animal can maintain its normal body temperature without needing to increase metabolic activity [178]. Newborn calves, although equipped with basic thermoregulatory abilities [179], are particularly vulnerable to cold stress. This is due to their high surface/mass ratio and poor insulation, including thin skin and limited subcutaneous fat [180,181]. Additionally, the heat generation of dystocial calves is often impaired, with thermogenesis reported to be up to 36% lower than in eutocial calves [162]. The lower critical temperature in calves decreases with age, from 13 °C at 1 day old to 6.4 °C at 30 days old [182]. Temperatures below this critical limit increase metabolic demands, and when low temperatures are combined with restricted nutrition, calves may experience impaired immune function [162,183], leading to increased susceptibility to diseases. Additionally, bedding moisture and the presence of drafts are considered significant risk factors for cold stress in calves [183,184]. For example, Lago et al. [185] reported a reduction in the prevalence of calf respiratory diseases with an increased nesting score (where the calf’s legs are completely covered by bedding material when lying down). This effect may be attributed to reduced maintenance energy requirement and improved immune system performance.

The upper critical temperature in pre-weaning calves is 26 °C [182], although other studies have identified varying thresholds for increased evaporative heat dissipation [186]. The temperature–humidity index (THI), considered the best environmental indicator of heat stress for calves [187], is reported to be 69 in subtropical conditions [187] and 78 in temperate climate conditions [188]. Several studies confirm that calves experience increased stress levels due to heat exposure. Calves increased average respiratory rates by approximately 50% [186], increased rectal temperature up to 39.7 °C [189] or 40.4 °C [190], increased cortisol concentrations [191,192], and showed altered behaviour such as reduced activity and increased shade-seeking behaviour [162,193].

### 8.3. Stocking Density

Several studies have concluded that a high stocking density negatively affects calf performance and health [106]. Calf stocking density was identified as a risk factor for diarrhea and respiratory problems [11]. For instance, having a limited space per calf (≤1.8 m^2^/calf) increased the odds of respiratory disease (OR: 2.13) compared to more spacious conditions (>1.8 m^2^/calf). Additionally, calves housed in individual hutches with more space (3.71 m^2^/calf) exhibited better pulmonary immunity, showing less eosinophil infiltration. Increased space availability allows calves to be more active and display natural play behaviour, and reduces stress-related behaviours such as non-nutritive oral activity [194]. Additionally, larger pen areas may help limit the spread of pathogens. A low stocking density, when combined with good ventilation, optimal temperature, low ammonia levels (<5.25 mg/m^3^), and controlled humidity, contributes to better air quality and lower prevalence of respiratory diseases.

For instance, appropriate ventilation and low stocking density have been linked to lower concentrations of airborne bacteria [185]. Similarly, keeping the pen or hutch temperature within the thermoneutral zone and maintaining low ammonia levels (i.e., <5.25 mg/m^3^; [195]) have been shown to reduce the prevalence of bovine respiratory disease (BRD) [196,197].

### 8.4. Practical Recommendations on Housing

It is recommended that calves are housed in stable pairs or small groups (2–7 calves) from the first week of life. In group housing, competition for milk can be mitigated by ensuring that the number of teats is equal to or greater than the number of calves in the group [198,199]. Additionally, the use of long feeding barriers that occupy the front half of the calf during feeding was shown to reduce competitive interactions [200]. To prevent cross-suckling, milk can be offered with slow-flow nipples [201] or with anti-sucking devices [202]. The implementation of precision livestock farming technologies such as automatic milk feeders and remote monitoring systems can further support social housing by enabling individualized care and reducing labour demand, even in group settings [27,203].

Housing can help reduce the negative effects of climate by allowing for a low stocking density, good ventilation, and low levels of ammonia in order to maintain calves in their thermoneutral zone.

Calves should be housed in a warm and dry environment. It is recommended to remove soiled bedding and manure and add fresh bedding so that calves can remain clean and dry [117]. Bedding materials should have a high ability to absorb water, provide insulation, and allow for ideal nesting conditions. Long wheat straw bedding [185,204] is thus preferable to fine particles of sawdust or sand.

In cold conditions, providing shelter from drafts and rain is particularly important. In indoor conditions, infrared heat lamps can be used to create a warm microclimate and are much more useful than calf jackets [205]. In hot conditions, shelter also can help reduce exposure to extreme solar radiation and the provision of shade and easily accessible drinking water is essential. In indoor conditions, further heat-stress mitigation, such as active forced ventilation, may be warranted.

### 8.5. Main Research Gaps on Housing

Further research is needed to fully determine the optimal housing systems for calves under different environmental and management conditions. Key research areas include the effects of the calf’s age when separated from the dam on stress, growth, and immune function, the appropriate age for grouping with conspecifics after birth, and the effects of group size. Addressing these gaps will facilitate the adoption of housing practices that are both welfare-friendly and economically efficient.

## 9. Diseases

In pre-weaned calves, the most important and prevalent gastroenteric disorders are grouped under the name of neonatal calf diarrhea (NCD) or calf scours [103]. NCD in calves is characterized by the acute appearance of loose or watery feces. The cause of diarrhea can be infectious or nutritional. NCD can affect calves from a few days of age (mainly *E. coli* K99+ or ETEC) to up to 1–2 months [11]. It affects between 10 and 35% of suckling calves and is responsible for more than 55% of pre-weaning losses [65,206,207,208]. NCD in calves induce sickness behaviour, which includes physiological and behavioural changes in the affected animals. NCD can cause variable degrees of dehydration, metabolic acidosis, hypothermia, visceral pain (colic), apathy, and depression [207,209], and lead to death from dehydration and acidosis if not treated [11].

Calves with diarrhea show lethargy, somnolence, and loss of appetite and thirst, and they may be more reluctant than usual to approach the stockperson. Also, sick calves decrease their general physical activity and lie down for longer periods than normal, reduce their self-grooming behaviour, and interact less with other animals [210].

In calves suffering from diarrhea, food malabsorption increases the LCT, and they are thus more susceptible to cold temperatures. As a consequence, trembling is frequently observed. A change in resting behaviour has also been described, at least in small calves, and calves suffering from diarrhea are more frequently observed with their limbs under their body and their head resting to one side. This posture allows the animal to reduce its body surface and therefore decrease the loss of body heat. Sickness behaviour also involves a negative emotional state, including depression, pain, and anhedonia (loss of interest or reactivity to usually pleasant stimuli) [211,212]. Diarrhea can be accompanied by pain, and calves suffering from diarrhea adopt a pain-relieving posture, with a tucked up abdomen and the tail between the rear legs, whilst they are standing [142].

In pre-weaned calves, respiratory disorders are second to gastroenteric disorder in terms of morbidity and mortality. Affected animals generally present with mucopurulent nasal secretion, dyspnea, coughing, lack of appetite, dullness, and fever (>39.5 °C) [213]. From birth to weaning, this affects around 20% of calves and is responsible for up to 14% of preweaning deaths [65,214].

Respiratory disorders can lead to negative emotional and physical states such as discomfort, pain, air hunger, and distress, primarily due to impaired lungs or airway function or damage. It has been demonstrated that stress significantly disrupt the viral–bacterial balance, contributing to the development of fatal BRD [215,216]. Clinical signs of BRD may vary from mild or even unapparent to very severe symptoms. When diagnosed or mistreated late, BRD may progress to chronic pneumonia, which can lead to suffering. Death can occur after several days but may also be sudden [11].

Bull et al. [217] identified different behaviours associated with respiratory disease in calves, indicative of sickness behaviour. Calves with respiratory disease show decreased feeding behaviour (e.g., reduced number of feeder visits, reduced intake, and decreased rumination) and decreased activity (e.g., decreased number of steps or increased lying times) [218]. In addition, exploratory and social behaviour are reduced. For instance, calves with respiratory problems showed reduced social grooming [218] and increased time isolated from the group [219,220].

Navel cord remnants (defined as the external remnants of the umbilical cord post-rupture up to the junction with skin [221]) can allow bacteria to enter the body. If antiseptic treatment is not applied and environmental hygiene is poor, this can lead to localized or systemic infection [17]. Navel infection is characterized by pain, heat, swelling, or purulent discharge. If calves develop navel stump abnormalities, they are at increased risk of negative health outcomes and decreased growth [222,223]. Between 1 and 29% of calves develop navel infection over the first few weeks of life [167]. Male calves have higher prevalence of abnormal navel scores than females calves [224]. This may partly be due to differences in how male and female calves are cared for [225,226]. However, anatomical differences might also lead to more soiling of the navel stump and cord remnants following urination in males, increasing infection risk [221].

Arthritis in calves, particularly septic arthritis, involves joint inflammation due to bacterial infection and is most prevalent in calves under 8 weeks old. This condition is often secondary to systemic infections such as omphalitis, pneumonia, or bacterial enteritis. Arthritis causes reduced appetite and lethargy, pain and lameness, and fever due to systemic infection. The most commonly affected joints include the carpus (front knee), stifle, and hock [227,228].

### 9.1. Practical Recommendations on Disease Prevention

First of all, it is essential to avoid stressful events, or at least the simultaneous occurrence of several stressors, as certain common stressful practices increase the risk of health disorders. Some examples of common stressful practices are separation from dam, poor handling, transport, and mutilations such as disbudding and weaning [11,215,216].

Disease prevention is crucial and more effective than treatment. It is recommended to implement several preventive measures, such as vaccination, good colostrum management (see colostrum management section), biosecurity, and sanitation in order to minimize the impact of infections in calves (see [30,31] for further review).

Vaccination against common viral infections can reduce the incidence and severity of clinical signs and improve the overall health and productivity of herds [229]. For instance, the vaccination of pregnant cows increases the number of viral-specific antibodies in the colostrum, which can protect calves against viral infection [102,230]. However, vaccines do not fix the problem unless they are used in combination with other prevention measures [231].

Strict biosecurity measures should be implemented on cattle farms as they mitigate the introduction of new diseases from external sources and reduce the transmission of infectious diseases within the farm [232,233]. The main aspects of biosecurity measures are the selection of purchased animals, isolation of purchased (quarantine) or sick animals, movement control (including all vehicles, animals, and human traffic), and sanitation. Employees should be trained in good practices, such as the principles of hygiene and disease security, and should work with younger animals before working on older animals to prevent the spread of disease. The presence of other animals (e.g., dogs, pigeons) and of people from outside the farm should be avoid as much as possible [234].

Sanitation covers the disinfection of materials entering the farm, using clean overalls during farm visits, and washing hands before and after working with sick or young animals [232]. Several studies have emphasized the importance of cleaning artificial nipples and buckets [117]. Cleaning before disinfecting is crucial to ensure that organic matter is removed, preventing it from inactivating the disinfectants during the disinfection process [30,31]. For instance, both chlorine-based disinfectants and hydrogen peroxide offer effective means of decontaminating surfaces and environments contaminated with *Bovine coronavirus* [235,236].

It is recommended to wash and disinfect pens in-between calves [102,209] and to clean the calf’s bed daily. Longer intervals between the renewal of bedding material are correlated with higher concentrations of harmful microorganisms in the air, which negatively affect calf disease and mortality rates [237]. Cleaning the calf’s bedding only 2–3 times/week or once/week were associated with increased odds of diarrhea (OR 4.94 and 11.89, respectively) compared to daily cleaning [106]. Similarly, calves born in filthy or contaminated calving pens may be easily infected by pathogenic *E. coli*, predisposing them to infectious diarrhea in the first day of life [102]. Individual calving pens, which are cleaned between each calving, have been associated with reduced disease risk in calves [167,234].

Air quality in calf housing should be monitored in order to reduce respiratory problems. A threshold of 10 ppm ammonia was recommended for cattle by the EFSA panel on Animal Health and Animal Welfare [238] and confirmed in recent papers (e.g., [239,240]). It is also recommended to increase ventilation rates (e.g., using positive-pressure air-distribution systems, without creating drafts) and to increase the pen area as well as the number of open sides of calf pens in order to reduce airborne bacterial counts [185]. It is also suggested to avoid completely closed barns with fully slatted floors [241].

Early diagnosis and prompt management of infected animals are crucial to limit the spread of infection diseases [30,31]. The routine (twice daily) monitoring of calves will increase the early detection of sick animals, allow for prompt treatment, and reduce the risk of disease outbreaks [242].

### 9.2. Main Research Gaps on Calf Diseases

Further research is needed to better understand the epidemiology and characterization of viral and bacterial infections in dairy calves [30,31]. Potential solutions involve conducting large-scale epidemiological studies, including prevalence monitoring and assessing distribution and transmission dynamics, which are essential for the successful implementation of management strategies. In parallel, regular monitoring programmes should be implemented to track infection trends, enabling timely and targeted interventions.

Although gastroenteric and respiratory problems in calves are widely studied, navel healing and navel care remain understudied. Addressing this gap requires the development of standardized scoring systems to assess navel healing, and an investigation into the effects of nutrition, housing, bedding, and management on navel healing outcomes. This will support the development of evidence-based guidelines for navel care, identifying the most effective preventive hygiene protocols and the best practices to enhance calf health and welfare.

## 10. Painful Husbandry Procedures

Disbudding or dehorning and castration are routine practices in cattle. Disbudding is the removal of horn-forming tissue before attachment to the skull, whereas dehorning is the removal of the horn after this occurs, typically at 2 to 3 months of age. Castration in males is the surgical or chemical removal or inactivation of the testes.

These procedures are commonly implemented to prevent issues related to animal welfare. Dehorning decreases the risk of injury to both people and other animals [86,243]. Polled animals also require less space in the pen and at the feeder than horned animals [243,244]. Males are routinely castrated to reduce aggressive and sexual behaviour and to achieve more desirable carcass characteristics [243].

The most commonly used procedures for disbudding are cauterization (hot iron) between 3 and 8 weeks of age and chemical disbudding (through the application of a caustic paste) before 3 weeks of age [86,243,245]. Animals above 3 months of age typically require surgical amputation dehorning [86,243,246]. The most common methods of castration are those in which the testicles are either removed (surgery), or killed by crushing (clamp) or constricting (rubber rings or latex bands) the tissues that supply blood to the testes [247].

Although these husbandry practices are justified for handling reasons, and even on animal welfare grounds, they cause pain in absence of appropriate pain management techniques [248]. All methods of disbudding, dehorning, and castration are painful and stressful at any age [249,250,251]. These procedures cause substantial behavioural and physiological responses when performed without pain relief, and these changes can persist for at least 24–48 h after disbudding and several weeks after castration [243,252]. Recent research has also shown that the wound remains sensitive to mechanical stimulation from 4 to 13 weeks [243,253].

In addition to causing pain, these procedures—if performed without the use of pain mitigation—induce a negative affective state in calves [57,254] and are strongly aversive [255]. More importantly, untreated early-life pain has been shown to disrupt brain development, with long-lasting behavioural implications [243,256]. This has been tested in different painful procedures and different farm animals. For example, there is some evidence that castrating soon after birth increases sensitivity to noxious stimuli later in life and it has been shown that lambs castrated at 1 day of age had greater pain responses to tail docking 1 month later than those castrated at 10 days of age [257].

### 10.1. Practical Recommendations on Painful Husbandry Procedures

To minimize pain or discomfort associated with routine husbandry procedures, key factors to consider include the animal’s age, the chosen method, appropriate medication, and the level of handler training [86,249].

Disbudding and castration should be performed at the youngest age possible to minimize tissue damage and improve ease of handling [243]. Disbudding is recommended over dehorning, because it is less invasive and less painful [249,252]. Disbudding can be performed as soon as the buds can be palpated (3–4 weeks old). Hot-iron disbudding is preferred to chemical disbudding [252], although published comparisons were presented using an uncontrolled volume of paste, instead of paste sticks.

All methods of castration are known to cause pain, but the evidence reviewed below indicates that the constriction methods (rubber ring and latex band) are the most problematic [247]. For instance, using the clamp method at an early age is preferable to rubber ring castration, as rubber bands causes more inflammation and the most persistent physiological and behavioural changes due to pain [253,258].

A multimodal pain relief combining local anesthesia and systemic analgesia using non-steroidal anti-inflammatory drugs (NSAIDs) should be used in order to minimize behavioural and hormonal changes that indicate acute pain [245,259]. Moreover, it is essential to consider extending the analgesic treatment for at least 24 h–48 h after the procedure [260]. Currently, there are no licenced drugs for managing long-term pain in farm animals in order to abolish behavioural changes and pain sensitivity in the following days and weeks after procedure [243,253,261].

To facilitate both the handling of calves and the surgical procedures, a sedative such as xylazine can be used in combination with multimodal pain relief. It is important to remember that sedatives cause muscle relaxation and limit the animals’ ability to move or react to stimuli, but sedation alone does not control pain [262].

The use of polled cattle must be considered as an alternative to dehorning in future. The presence of horns is a recessive trait in cattle, requiring both parents to have a horned gene to continue the trait [263].

Immunocastration, which involves vaccination against the GnRH-I hormone, is an effective alternative to surgical castration in controlling reproduction and promoting animal welfare in several species, including beef calves [264].

### 10.2. Main Research Gaps on Painful Husbandry Procedures

Further research is required to quantify the pain associated with the use of caustic paste applied via sticks.

Greater attention should be given to the long-term impacts of painful husbandry procedures on calf welfare and growth. The current research primarily focuses on immediate or short-term pain relief, leaving calves vulnerable to prolonged pain and increased pain sensitivity, sometimes lasting for days or even weeks following a procedure. Potential solutions include investigating long-acting analgesics or multimodal pain management strategies to mitigate persistent pain. These studies should be adapted to different procedures and calf age groups to establish standardized protocols suitable for practical field application.

There is growing interest in alternatives to traditional husbandry procedures, such as immunocastration or the use of polled calves. However, further research on the cost-effectiveness and consumer acceptability of these alternatives under commercial farming conditions is needed.

## 11. Transport

With large variations depending on geography, calves may be transported over long distances. Journeys may go directly from the farm of origin to a rearing or a fattening farm but can also involve various combinations of markets, assembly centres, and stops at control posts [265].

All types of transport are considered multifactorial stressors. They involve several sources of stress, including group stress, handling, motion stress, restriction of movement, difficulty resting, and sensory overstimulation [265,266]. Additionally, long-distance transportation also involves prolonged hunger and thirst. As a result, transport can lead to severe negative animal welfare consequences, such as boredom, discomfort, stress and distress, fatigue, fear, frustration, and pain [265,267].

In addition, these hazards occur at an age when calves are still immature and several of their physiological systems are still developing. Young calves, for example, are still developing their gastrointestinal tract, and their thermoregulatory and acquired immune systems are not yet fully functional [265,268]. Stress can also compromise the calves’ immune systems, making them more susceptible to diseases [269], such as bovine respiratory disease (BRD) [265,270].

### 11.1. Practical Recommendations on Transport of Unweaned Calves

Preventive measures and practical recommendations for the transport of unweaned calves are the same as those for adult cattle transported by road. Several general recommendations for cattle transport—covering the preparation phase, loading and unloading, and control post stops for long-distance transportation—are reviewed in the EFSA Scientific Opinion ([265]).

Additionally, unweaned calves are more difficult to handle than adult cattle because they do not exhibit natural herding or group behaviour [265]. As a result, having well-trained staff (including veterinarians, farmers, and livestock drivers), as well as ensuring the proper maintenance of facilities, are key factors to consider.

Avoiding extreme thermal conditions during transportation is essential [222], as the thermoneutral zone for unweaned calves is estimated to be between 15 and 25 °C. Providing proper bedding helps to reduce not only the risk of cold stress but also resting discomfort during transport [265].

Calves prefer to lie down during transport [269]. To allow them to do so comfortably, a minimum amount of floor per animal must be provided. This can be estimated using the formula A = K × W^2/3^ (where A is area in m^2^ per animal, K is a constant, and W is live weight in kg). Research indicates that calves require at least k = 0.027 to have sufficient space to lie down during journeys [265].

Calves with an infected or wet navel, low BW, and/or apathetic behaviour should not be transported, as their welfare will be more compromised, increasing the likelihood of negative health outcomes [271].

To prevent prolonged hunger, evidence suggests that calves should be fed a milk meal within the 12 h prior to transport. They also require at least 3 h of rest for proper digestion, meaning a 1 h journey break is insufficient to meet their feeding and resting needs [265].

When calves are prepared for transport at an assembly centre, they may be given an electrolyte solution (or a glucose–electrolyte mix) as their last meal. However, this does not meet their nutritional needs and cannot replace a milk or milk replacer meal [265].

It has been suggested to administer an NSAIDs before long-distance transportation, as this can reduce navel inflammation and rectal temperature upon arrival and increase feed intake and growth afterward [266,272].

### 11.2. Main Research Gaps on Transport of Unweaned Calves

There is a lack of consensus on how to improve calf welfare during transportation, especially long-distance transport. Further research is needed to clarify the effects of transportation on affective states and natural behaviour in young calves. Potential solutions involve developing evidence-based guidelines for transport duration, rest periods, and handling procedures.

## 12. Concluding Remarks

This review demonstrates that calf welfare is closely linked to husbandry practices and health conditions during early life. Poor management of colostrum and milk feeding can result in failure of passive immunity transfer, hunger, impaired growth, and increased disease susceptibility. Housing conditions—especially space allowance, group size, and bedding—substantially influence calf behaviour, thermal comfort, and pathogen exposure. Health disorders such as neonatal calf diarrhea and bovine respiratory disease remain the leading causes of neonatal morbidity and mortality, with direct implications for welfare. Painful procedures like disbudding and castration, if performed without analgesia, are significant welfare concerns. Stressors related to transport, early separation from the dam, and abrupt weaning also negatively affect calf welfare.

Current evidence supports several key recommendations, highlighting practical management strategies. Around calving, maternity pens should be used to allow cows to isolate themselves and to enable the dam to lick the calf. Staff should be trained to recognize prolonged or abnormal parturition and to effectively assist in calving without causing additional pain. It is recommended to provide 10–12% of BW in colostrum within 2–3 h after birth, followed by 5% of BW at 6–12 h, and transition milk for the first 4 days. Refractometers should be used to ensure high-quality colostrum (serum total protein > 52 g/L or Brix value > 9.4%), and oesophageal tubes should only be used for weak or non-suckling calves. Milk feeding should be provided at approximately 20% BW during the first 4 weeks, divided into at least two meals per day. Only high-quality milk replacer or whole milk should be used, avoiding waste milk. Both colostrum and milk should be administered via a nipple-feeding system, and all equipment must be cleaned and disinfected after each use. Clean water should be provided from birth (or by 2–3 days of age) to support hydration and rumen development. Weaning should not be early or abrupt; instead, a gradual step-down reduction in milk over a minimum of 10 days is recommended. Calves should consume at least 1 kg of solid feed per day for three consecutive days before weaning.

From the first week of life, calves should be housed in stable pairs or small groups (2–7 animals). To prevent competition in group housing, it is important to provide an equal or greater number of teats compared to calves, long feeding barriers, and slow-flow or anti-suckling devices. Housing should maintain a thermoneutral environment (13 °C to 26 °C at one day of age), remain dry and well-ventilated, and include bedding that absorbs moisture and provides insulation. Vaccination programmes for both dams and calves should be implemented, along with biosecurity measures such as quarantining new animals and controlling visitor and vehicle access. Painful husbandry procedures (e.g., disbudding, castration) should always include effective multimodal analgesia, and alternatives such as polled breeding or immunocastration may be considered. Weak, wet, low-BW, or navel-infected calves should not be transported. During transport, calves must be kept within their thermoneutral zone, provided with bedding and sufficient lying space, and, on long journeys, given milk every 12 h. It is important to note that these recommendations are primarily directed towards intensive farming systems, and their applicability in semi-extensive or extensive systems may not always be feasible or scientifically justified.

However, important knowledge gaps persist. This review indicates that improving calf welfare requires four complementary strategies. First, controlled studies are needed to fill key knowledge gaps, including calving management, cow–calf separation, colostrum immune components, the treatment of long-lasting pain, and weaning strategies, among others. Second, standardized protocols and guidelines should be developed for areas where inconsistent practices currently limit welfare outcomes, such as perinatal mortality, colostrum quality thresholds, milk feeding, weaning, disease prevention, and transport. Third, tools and technologies should be validated, for instance, on-farm tests for colostrum quality and water quality or sensors to monitor stress and health related to weaning or transport. And, finally, the training of and knowledge transfer to farmers and veterinarians are essential to ensure consistent data collection, the adoption of best practices, and the implementation of evidence-based recommendations.

## Figures and Tables

**Table 1 animals-15-03064-t001:** Keywords, inclusion, and exclusion criteria used for database searches.

Databases		Web of Science and Scopus
Searches		Title, abstract or keywords
Dates		From 1990 to 08/2025
General Search terms		(calf OR calves) AND ((welfare OR wellbeing OR well-being) OR (management OR ‘husbandry practice*’ OR health OR disease* OR infecti* OR disorder* OR mortality))
Specific Search terms		
Topic	Subtopic	
Assisted Calving	Calving	AND (calving OR parturition OR dystocia OR ‘assist* calving’ OR ‘assist* parturition’)
	Perinatal mortality	AND (‘perinatal mortality’ OR stillbirth)
	Cow-calf separation	AND (‘early separat*’ OR mother OR separat* OR ‘maternal bond’)
Colostrum management		AND (colostrum* OR ‘transition milk’ OR ‘failed transfer passive immunity’)
Milk feeding management		AND (‘milk feed*’ OR ‘milk bucket’ OR bottle OR nipple OR ‘waste milk’)
Water provision		AND (drink* OR thirst)
Weaning management		AND (wean* OR ‘solid feed’ OR starter)
Housing	Social housing	AND (housing OR ‘pair hous*’ OR ‘social group’)
	Thermal stress	AND (‘cold stress’ OR ‘heat stress’)
	Stocking density	AND (‘stocking density’ OR ‘stocking rate’ OR ‘space allowance’ OR density OR overcrowd*)
Diseases	Diarrhea	AND (Diarrhoea OR diarrhea OR scour OR ncd) AND pain
	Respiratory disorders	AND (respiratory OR brd OR pneumonia OR cough* OR ‘ocular discharge’ OR ‘nasal discharge’)
	Omphalitis	AND (omphalitis OR navel)
	Arthritis	AND (arthritis OR lameness) AND pain
Painful husbandry procedures	Disbudding	AND (disbud* OR dehorn*) AND (pain OR discomfort OR sickness behav*)
	Castration	AND (castrat*) AND (pain OR discomfort OR sickness behav *)
Transport		AND (transport* OR *loading OR journey)
Inclusion criteria		Focused on calves
		Peer-reviewed articles (including primary research, reviews and meta-analyses), books or book chapters or EFSA scientific opinions
		Published or translated in English
		Studies focused on the impacts of husbandry practices and/or health condition on calf welfare
Exclusion criteria		Focused on other species or other cattle categories
		Not in English
		Duplicates
		Abstract only
		Studies that did not focus on the impacts of husbandry practices and/or health conditions on calf welfare

* indicates a truncation symbol, and ‘’ indicates an exact phrase, used in database searches.

## Data Availability

The original contributions presented in this study are included in the article. Further inquiries can be directed to the corresponding author.

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
