# Peer review of "Main Husbandry Practices and Health Conditions That Affect Welfare in Calves: A Narrative Review"

_animals, 2025, doi:10.3390/ani15213064_

Round 1

Reviewer 1 Report

Comments and Suggestions for Authors

General Comments

Clearly specify the original contribution of the work.

There is no description of how articles were selected, nor of databases, inclusion/exclusion criteria, or analysis methods.

Gaps are presented broadly, without detailing their relevance or potential solutions.

Long and repetitive sentences, particularly in the abstract and conclusion, I suggest making the text more concise and direct.

The text contains extensive blocks, making reading and comprehension difficult.

Transitions between topics are not smooth.

Conclusions could be more practically applicable, including management solutions or tools.

I suggest including tables comparing study results.

Abstract

Lack of specificity; the abstract is generic.

Methodology not described; even for a narrative review, briefly mentioning databases, inclusion/exclusion criteria, and methods increases credibility.

Gaps are presented broadly, without detailing practical relevance or potential solutions.

Some ideas are repeated (best practices, biosecurity, housing, and weaning).

Long and complex sentences reduce readability.

1.Introduction

The relationship between welfare, immunity, and diseases is mentioned but not detailed; physiological mechanisms and specific examples could be explored.

Little differentiation between production systems; reported issues in the EU do not distinguish between intensive, semi-intensive, or extensive systems.

The need for the current narrative review could be more clearly justified.

  1. Materials and Methods

Lack of detail on inclusion/exclusion criteria.

Search terms are broad and not refined, which may generate irrelevant results or omit important studies.

Reports and technical sheets are poorly described; it should be clarified which documents were included and how they were scientifically evaluated.

No mention of reference management software or strategies; including this would improve reproducibility.

  1. Calving and Assisted Calving (Dystocia)

Sentences are long and dense, making reading difficult.

Information on mortality, pain, and dystocia is repeated.

I suggest shorter titles.

I suggest improving transitions between paragraphs.

Clearly differentiate evidence from practical recommendations.

  1. Colostrum Management

Long and dense sentences combining different concepts (quality, quantity, absorption, disease risk).

Transitions between topics are not smooth.

Subtitles are long and not concise.

Limited quantitative approach; comparative tables could strengthen recommendations.

Practical and ethical aspects are poorly integrated; protocols to minimize stress or risk to calves could be detailed.

  1. Milk Feeding Management

Long sentences combining behavioral, physiological, and nutritional effects; clarity is reduced.

Repetition of information on volume, frequency, feeding method, and health effects.

Transitions between topics are not smooth; the sequence could be reorganized (quantity → frequency → method → milk type → hygiene → health effects).

Long subtitles; could be more concise.

Mixing of scientific evidence and practical recommendations without a clear distinction.

  1. Water Provision

Long and dense sentences combine physiological, behavioral, and environmental effects.

Repetition of concepts regarding the importance of water, fiber intake, and health development.

Mixing study results, recommendations, and management justifications.

It would be useful to conclude with a synthesis connecting water availability, quality, and calf welfare.

  1. Housing

Long sentences with multiple ideas make reading difficult.

Information on behavior, health, and management appears mixed; flow is confusing.

Text is very descriptive; including summary tables of practical recommendations would be helpful.

Some information is repetitive (microclimate, stocking density).

Shorter sentences are recommended; avoid repetition and integrate evidence with recommendations.

  1. Diseases

Paragraphs are very long; it is recommended to divide them by a central idea.

An extensive listing of clinical/behavioral signs disrupts readability.

Biosecurity, hygiene, and cleanliness are repeated; consolidate them in a single, organized paragraph.

Many prevalence data are presented, but without comparisons between diseases or regions.

  1. Painful Husbandry Procedures

Long text with repeated mentions of pain and absence of established strategies; it is recommended to synthesize.

Paragraphs mix management justification, method description, and consequences, reducing clarity.

Some sentences are overly interpretative (“undoubtedly painful practices”).

Excessively long paragraphs hinder reading.

  1. Transport

Text repeats that transport is harmful and that young calves are more vulnerable.

Long and unclear enumeration of stress factors.

The formula is presented without adequate explanation; better contextualization is recommended.

  1. Concluding Remarks

Sentences are very broad; terms such as “however,” “moreover,” and “remains unclear” appear repeatedly.

Gaps and recommendations are presented broadly; it would be helpful to indicate priority areas and suggested methodologies.

Alternatives for risky transport are mentioned generically; including already studied options would be useful.

Practical recommendations are presented without discussing adoption limitations in different systems.

Author Response

General Comments

R: Clearly specify the original contribution of the work.

AU: Although several reviews have focused on specific issues of calf welfare (e.g., colostrum), there is no comprehensive summary of the most important management practices and health issues in calves in relation to animal welfare. The last paper addressing this topic was published in 2008. In our review, we highlight the animal welfare issues that have emerged and the developments in the sector over the past 17 years, in order to better clarify our contribution (see L109-113).

There is no description of how articles were selected, nor of databases, inclusion/exclusion criteria, or analysis methods.

AU: Agreed, additional information on the specific keywords, inclusion and exclusion criteria, and the software used has been included. (see Table 1 and L121-125 and L135-139)

R: Gaps are presented broadly, without detailing their relevance or potential solutions.

AU: Agreed, gaps sections have been rewriting, aiming to detail practical relevance and specific potential solutions. See L212-234 (assisted calving), L310-327 (colostrum), L431-435 (milk feeding), L475-485 (water), L550-L552 (weaning), L691-699 (housing), L856-870 (diseases), L950-968 (painful husbandry procedures) and L1036-1038 (transport).

R: Long and repetitive sentences, particularly in the abstract and conclusion, I suggest making the text more concise and direct.

AU: Agreed. We have avoided repetitions and long sentences throughout the manuscript, with particular focus on the abstract and conclusions.

R: The text contains extensive blocks, making reading and comprehension difficult.

AU: Agreed. We have deleted some redundant information, and extensive paragraphs have been reduced throughout the manuscript.

R: Transitions between topics are not smooth.

AU: Agreed. We have improved the connections between topics throughout the manuscript.

R: Conclusions could be more practically applicable, including management solutions or tools.

AU: Recommendations have been included in greater detail, highlighting practical management strategies should be implemented (see L1052-1081)

R: I suggest including tables comparing study results.

AU: We agree that comparative tables are valuable in systematic reviews. However, our work is a narrative review aiming to provide a broad overview of management practices, health issues, and emerging welfare concerns. The studies we reviewed are highly heterogeneous in design and outcomes, which makes direct comparison unfeasible. Instead, we refer to comparative tables published in previous systematic reviews (e.g. see L261)

Abstract

R: Lack of specificity; the abstract is generic.

AU: Agreed. We have increased the specificity in the abstract.

R: Methodology not described; even for a narrative review, briefly mentioning databases, inclusion/exclusion criteria, and methods increases credibility.

AU: Agreed. The basic methodology is mentioned (see L33–L35).

R: Gaps are presented broadly, without detailing practical relevance or potential solutions.

AU: Agreed. The gaps in the abstract have been detailed (see L46–L49).

R: Some ideas are repeated (best practices, biosecurity, housing, and weaning).

AU: Agreed. Repetitions have been removed.

R: Long and complex sentences reduce readability.

AU: Agreed. Sentences in the abstract (and throughout the manuscript) have been reduced to improve readability.

AU (abstract in general): More detailed information in response to the reviewer’s comments cannot be included in the abstract, as the maximum word count allowed is 200 words.

1.Introduction

R: The relationship between welfare, immunity, and diseases is mentioned but not detailed; physiological mechanisms and specific examples could be explored.

AU: We’ve detailed the relationship between the effects of the two stages of the immune system and welfare (see L70-82). Specific examples have been described later in the manuscript (for instance, sickness behaviour see L708, L722, L754)

R: Little differentiation between production systems; reported issues in the EU do not distinguish between intensive, semi-intensive, or extensive systems.

AU: The revision has focused on intensive farms. We’ve included this clarification (see L92). Other welfare problems, which are only related to extensive farms (such as parasites common in pasture), have not been included. This limitation has also included in the conclusion (see L1081-1084)

R: The need for the current narrative review could be more clearly justified.

AU: Based on our search, the last review summarizing the main management practices and health issues related to animal welfare was published in 2008 (Stull and Reynolds). We have added the most important new developments from the past 17 years and emphasized the need to update this information in a single comprehensive document (see L109-L113).

Materials and Methods

R: Lack of detail on inclusion/exclusion criteria.

AU: Details of the inclusion and exclusion criteria are presented in Table 1

R: Search terms are broad and not refined, which may generate irrelevant results or omit important studies.

AU: The specific search terms have been presented in Table 1.

R: Reports and technical sheets are poorly described; it should be clarified which documents were included and how they were scientifically evaluated.

AU: In order to avoid doubts about the scientific reliability of the information provided, we decided to include only peer-reviewed papers, books, and scientific EFSA opinions (see L121–125 and Table1). Technical sheets previously included in the manuscript (mainly from universities and European projects) have been deleted: Care4Dairy project (reference nº78 deleted), fact sheet of Salfer, 2000 (reference nº207 deleted), NAHMS fact sheet (reference nº210 deleted) and FAWEC fact sheet (reference nº215 deleted). In consequences, the information has been corrected accordingly (by changing the reference or slightly modifying the text in cases where no scientific evidence is available).

R: No mention of reference management software or strategies; including this would improve reproducibility.

AU: Agreed. The two software programs used (Tera-Tools and Zotero) have been included (see L136-139).

  1. Calving and Assisted Calving (Dystocia)

R: Sentences are long and dense, making reading difficult.

AU: Agreed, sentences have been shortened to improve readability

R: Information on mortality, pain, and dystocia is repeated.

AU: Agreed, redundant information related to mortality/pain has been removed

R: I suggest shorter titles.

AU: “Calving and assisted calving (dystocia)” has been changed to “assisted calving” and subtitle 3.1 has been shortened (Practical recommendations on assisted calving).

R: I suggest improving transitions between paragraphs.

AU: Agreed, transition between sentences and paragraphs have been added to improve logical flow

R: Clearly differentiate evidence from practical recommendations.

AU: Agreed. We have included scientific evidence in the “Assisted Calving” section and practical recommendations in the “Recommendations” section. Some specific recommendations are also supported by scientific evidence to clarify that they are well tested.

  1. Colostrum Management

R: Long and dense sentences combining different concepts (quality, quantity, absorption, disease risk).

AU: The sentences have been shortened. The 3 keys important factors for colostrum (quality, quantity and absorption) have been better structured/organized.

R: Transitions between topics are not smooth.

AU: Agreed, we have added transitions between paragraphs to improve the connections within the text.

R: Subtitles are long and not concise.

Subtitle 4.1 has been shortened (Practical recommendations on colostrum management).

R: Limited quantitative approach; comparative tables could strengthen recommendations.

AU: A specific quantitative approach to immunoglobulins in colostrum has been included in the recommendation section (see L289–290). The manuscript refers to a comparison table of different quantitative approaches (Table 2 in Carulla et al., 2023a) (see L261). We feel it is unnecessary to repeat this table, as it is already updated and concise.

R: Practical and ethical aspects are poorly integrated; protocols to minimize stress or risk to calves could be detailed.

AU: Specific recommendations are now detailed.

  1. Milk Feeding Management

R: Long sentences combining behavioral, physiological, and nutritional effects; clarity is reduced.

AU: Agreed. We have split long sentences to improve readability. Grouped behavioural, physiological, and nutritional effects into separate sentences. Removed redundancies (eg. “in consequence”)

R: Repetition of information on volume, frequency, feeding method, and health effects.

AU: We have removed repeated information on these topics. Some specific examples have been deleted and are now cited as references in the general information (e.g. L397-L400)

R: Transitions between topics are not smooth; the sequence could be reorganized (quantity → frequency → method → milk type → hygiene → health effects).

AU: Agreed. Information has been presented following your recommendation: (1) quantity, (2) frequency, (3) method, (4) milk type, (5) hygiene and (6) health effects

R: Long subtitles; could be more concise.

AU: Subtitle 5.1 has been shortened (Practical recommendations on milk feeding management)

R: Mixing of scientific evidence and practical recommendations without a clear distinction.

AU: As our recommendations are based on science, sometimes is difficult not to mix them. However, we have been presented the recommendations as precisely as possible.

  1. Water Provision

R: Long and dense sentences combine physiological, behavioral, and environmental effects.

AU: The text has been rewritten to reduce long and dense sentences.

R: Repetition of concepts regarding the importance of water, fiber intake, and health development.

AU: redundant information related to water quality has been deleted (see L465 and L473-475). Importance of water has been explained by two important factors: (1) quantity and (2) quality, which the consequences are not exactly the same. No repetition related to fiber intake has been detected in the water provision section.

R: Mixing study results, recommendations, and management justifications.

AU: We have focused on specific recommendations.

R: It would be useful to conclude with a synthesis connecting water availability, quality, and calf welfare.

AU: Agreed, see L480-481.

AU: Although there were no comments regarding the weaning section, we have revised the text by reducing long sentences, improving transitions between paragraphs, and focusing on specific gaps and recommendations, similarly to the rest of the manuscript.

  1. Housing

R: Long sentences with multiple ideas make reading difficult.

AU: Agreed. The long sentences have been split.

R: Information on behavior, health, and management appears mixed; flow is confusing.

AU: Agreed. The text has been rewriten in order to improve flow.

R: Text is very descriptive; including summary tables of practical recommendations would be helpful.

AU: Agreed. Descriptive sections have been removed.

R: Some information is repetitive (microclimate, stocking density).

AU: Agreed. Information regarding stocking density has been presented together.

R: Shorter sentences are recommended; avoid repetition and integrate evidence with recommendations.

AU: Agreed, the recommendations have been shortened. In some cases, supporting scientific evidence has been deleted

  1. Diseases

R: Paragraphs are very long; it is recommended to divide them by a central idea.

AU: Agreed. Sentences have been shortened and some paragraphs deleted.

R: An extensive listing of clinical/behavioral signs disrupts readability.

AU: The text has been rephrased in order to improve readability

R: Biosecurity, hygiene, and cleanliness are repeated; consolidate them in a single, organized paragraph.

AU: Some repetitions related to hygiene/cleanliness have been deleted. Three main aspects are covered: biosecurity, disinfection of materials and disinfection of equipment.

R: Many prevalence data are presented, but without comparisons between diseases or regions.

AU: Specific data related to pathogens have been retained, as pathogens are associated with specific diseases (e.g., E. coli – diarrhea). Comparison between regions is difficult and not the aim of the present review.

  1. Painful Husbandry Procedures

R: Long text with repeated mentions of pain and absence of established strategies; it is recommended to synthesize.

AU: Agreed. We have revised the text and deleted some paragraphs.

R: Paragraphs mix management justification, method description, and consequences, reducing clarity.

AU: The three sections (justification, method description and consequences) have been split and clearly differentiated.

R: Some sentences are overly interpretative (“undoubtedly painful practices”).

AU: Agreed, it has been deleted (see L901)

R: Excessively long paragraphs hinder reading.

AU: Agreed. We have revised the text and deleted some paragraphs.

  1. Transport

R: Text repeats that transport is harmful and that young calves are more vulnerable.

AU: Agreed. This concept has been deleted in one of the sections (see L1025-1027)

R: Long and unclear enumeration of stress factors.

AU: The list has been reduced and some terms clarified.

R: The formula is presented without adequate explanation; better contextualization is recommended.

AU: Agreed, the formula has been better explained (see L912-917)

  1. Concluding Remarks

R: Sentences are very broad; terms such as “however,” “moreover,” and “remains unclear” appear repeatedly.

R: Gaps and recommendations are presented broadly; it would be helpful to indicate priority areas and suggested methodologies.

AU: Four potential solutions to resolve the gaps have been identify and included in the concluding remarks (See L1102-1112). Recommendations have been included in greater detail, highlighting practical management strategies should be implemented (see L1062-1093).

R: Alternatives for risky transport are mentioned generically; including already studied options would be useful.

AU: Specific recommendations or factors to be considered during calf transport are detailed in L1088–1090.

R: Practical recommendations are presented without discussing adoption limitations in different systems.

 AU: Agreed. We have specified that the review is focused on animal welfare issues in intensive production systems (see L92), and consequently, the recommendations are as well (see L1090-1093)

Reviewer 2 Report

Comments and Suggestions for Authors

The presented article is a comprehensive scientific paper that exhaustively analyzes the impact of key husbandry practices and health conditions on the welfare of calves during the crucial, early period of their lives. As a narrative review, it serves as an adequate and current presentation of knowledge on this topic

The authors systematically discuss the most important aspects of calf rearing, basing their conclusions on current scientific research findings. The paper's methodology is based on a review of peer-reviewed scientific articles from the Web of Science and Scopus databases, which ensures its high substantive level

The article is characterized by a two-pronged approach. On one hand, based on the synthesized knowledge, the authors formulate clear, evidence-based recommendations that can be directly implemented in farming practice to improve calf welfare. This compilation is extremely valuable for farmers, veterinarians, and advisors. On the other hand, the paper diligently points out areas requiring further research, identifying critical knowledge gaps. 

In conclusion, the article "Main husbandry practices and health conditions that affect welfare in calves: A narrative review" constitutes a valuable and complete source of knowledge on the factors shaping calf welfare. By combining a synthesis of current research, practical recommendations, and the identification of future research directions, this work is a valuable tool for both the scientific community and individuals directly involved in cattle farming.

Author Response

Thank you very much for your comments. We appreciate them greatly.

Following Reviewer 1’s suggestions, we made some changes, but the essence of the article remains the same. The main revisions include a clearer explanation of our literature search methodology and improvements to the overall readability of the text. In addition, the identified gaps and recommendations have been presented more directly.